# Endogenous Opioids in Crohn’s Disease

**DOI:** 10.3390/biomedicines11072037

**Published:** 2023-07-20

**Authors:** Adrian Martyniak, Andrzej Wędrychowicz, Przemysław J. Tomasik

**Affiliations:** 1Department of Clinical Biochemistry, Pediatric Institute, Faculty of Medicine, Jagiellonian University Medical College, 30-663 Krakow, Poland; adrian.martyniak@uj.edu.pl; 2Department of Pediatrics, Gastroenterology and Nutrition, Pediatric Institute, Faculty of Medicine, Jagiellonian University Medical College, 30-663 Krakow, Poland; andrzej.wedrychowicz@uj.edu.pl

**Keywords:** endogenous opioids, inflammatory bowel disease, endorphin, enkephalin, dynorphin

## Abstract

Caring for patients with Crohn’s disease (CD) is a serious challenge in modern medicine. The increasing incidence of CD among adolescents and the severe course of the disease create the need for new methods of diagnosis and therapy. Endogenous opioids are a group of low molecular weight chemical compounds with analgesic and anti-inflammatory properties. Endorphins, enkephalins, and dynorphins may have potentially beneficial effects on the course of CD. Previous research data on this topic are inconsistent. Some authors have reported an increase in the concentration of leukocytes during the course of inflammatory bowel disease (IBD) while others have described a downward trend, explained by DPP-IV enzyme activity. Even fewer data are available on plasma endo-opioid level. There is also a lack of comprehensive studies that have assessed the endo-opioid system in patients with IBD. Therefore, the objective of this study was to measure the serum concentrations of human β-endorphin, human proenkephalin (A), and human big dynorphin in CD patients in the acute phase of the disease, during hospital treatment, and in the remission state. All determinations were performed using ELISA kits. The results of our study showed that the concentrations of all the tested endo-opioids, especially β-endorphin and proenkephalin (A), were reduced in adolescents with CD compared to those in the healthy control group, during the acute phase of the disease, and in the remission state. Modulation of the endogenous opioid system and the use of selective nonnarcotic agonists of opioid receptors seems to be promising goals in the future treatment of CD.

## 1. Introduction

The incidence of Crohn’s disease (CD), depending on the region, ranges from 5 to 20.2/100,000 cases per year [1]. The highest incidence rates have been recorded in developed countries, especially in Europe and North America [2]. The highest incidence rate has been observed in the second and third decade of life, but about 25% of cases have been diagnosed in adolescents 10–19 years of age [3,4]. The actual theory on the pathogenesis of inflammatory bowel disease (IBD) suggests a loss of tolerance to intestinal microbiota and abnormal bacteria–host interactions, which stimulate an immune-mediated response and upregulate inflammation in the gut [5,6,7]. 

The most common symptoms of CD are abdominal pain and diarrhea [8]. Patients may complain of varied intensities of abdominal pain in different locations, while chronic diarrhea may provoke a failure to thrive and nutritional deficiencies [9]. These symptoms can cause a significant decrease in the quality of life of CD patients. 

Treatment of CD is a serious challenge. In addition to specific pharmacological treatments, opioids are still regarded to be the gold standard in the therapy of moderate to severe pain [10]. The chronic use of opioids to treat abdominal pain is ”alarmingly high” among patients with IBD [11]. The most common and severe side effects of chronic opioid use include inhibition of gastrointestinal motility (i.e., opioid-induced intestinal dysfunction and opioid-induced constipation), respiratory center depression, and physical dependence [12,13]. The constipating and antidiarrheal effects of exogenously administered opioids have been extensively documented [14]. Mainly, opioids inhibit fluid and electrolyte secretion and stimulate gut smooth muscle contractions; both effects are induced by interactions with different types of opioid receptors (ORs) located in the gut and the central nervous system [15]. 

Opioids are also synthesized in the human body; these compounds are called endogenous opioids or endo-opioids, which are a group of small-molecule peptides that consists of three main families, i.e., endorphins, dynorphins, and enkephalins [16]. Endoopioids are widely distributed in many tissues and organs and also synthesized in leukocytes, including CD4+ T lymphocytes and neutrophils [17,18,19]. In the gut, endorphins are found in some neuroendocrine cells, such as enterochromaffin cells in the gut, endocrine pancreatic cells, and gastrin cells in the antrum. Enkephalins and dynorphins colocalize with neuronal transmitters, for example, acetylcholine, substance P, vasoactive intestinal peptide (VIP), or galanin [20]. 

Under physiological conditions, endogenous opioids play many important functions. In the gut, they are responsible for the inhibition of gastric emptying and intestinal motility. They attenuate biliary, pancreatic, and intestinal secretions, and they exert immunomodulatory effects. Furthermore, they modulate intestinal bacterial microbiota by increasing intestinal permeability and facilitating bacterial translocation [21]. They have analgetic properties and regulate the stress response [22]. They affect mood, and they play a key role in the modulation of the brain’s reward center, which influences behavioral and social behavior [23,24]. The impact of endogenous opioids on the immune system can be twofold, i.e., immunosuppressive and immunomodulatory [25,26]. 

Endogenous and exogenous opioids act through specific opioid receptors that are located in the central and peripheral nervous system, the gastrointestinal tract, and the immune system [27]. Opioid receptors are divided into three types, namely µ-opioid receptors (MORs), δ-opioid receptors (DORs), and κ-opioid receptors (KORs), which belong to the G protein-coupled receptor (GPCR) family [28]. Each type of receptor possesses a different ligand-binding ability, and thus regulates distinct signaling pathways. β-Endorphin has a similar affinity to MOR and DOR receptors, dynorphins are considered KOR agonists, and enkephalins are the preferred ligands for DOR [29]. 

Endogenous opioid levels and their actions have been extensively studied in CD as a potential target for the treatment, among other minimizing exogenous opioid treatments [27].

Our research study aimed to assess the dynamic changes in the concentrations of β-endorphin, proenkephalin (PENK), and big dynorphin in the serum of adolescents with CD, divided into three groups according to the acute phase of the disease, during hospital treatment, and the remission state of CD.

## 2. Materials and Methods

### 2.1. Studied Groups

We recruited adolescents suffering from CD among the patients of the Department of Pediatrics, Gastroenterology and Nutrition, University Children’s Hospital in Krakow, Poland. The diagnosis of CD was based on the revised Porto criteria [30]. All CD patients were newly diagnosed and no patient had been previously treated with pharmacological or nutritional therapy before their enrollment in this study. In the CD patient group, fasting blood samples were taken three times: during the active phase of the disease (CD1) before starting hospital treatment, up to 24 h after admission to the ward; during the hospital treatment, i.e., 2–4 weeks later in the course of hospital care (CD2); and in the remission state of the disease, during outpatients control visits, 3–6 months after the hospital treatment (CD3). Adolescents in the control group were recruited from families and friends of the researchers of the study. These adolescents were without any pathological clinical signs and complaints and any pharmacological treatment. In the control group, fasting blood samples were drawn once. All samples were sent immediately to the hospital laboratory.

### 2.2. Laboratory Testing

The blood samples were centrifugated, and the separated serum was frozen at −80 centigrade until measurement. The maximum banking time was not longer than 14 months. The endogenous opioid concentrations were measured using commercially available enzyme-linked immunosorbent assay (ELISA) immunoassays: human β-endorphin, human proenkephalin (A), and human big dynorphin kits from Fine Test (Wuhan Fine Biotech Co., Ltd., Wuhan, China). The ranges of the ELISA kits and sensitivities were as follows: Human big dynorphin kit’s range of 7.813–500 pg/mL and sensitivity of 4.688 pg/mL, human beta endorphin kit’s range of 15.625–1000 pg/mL and sensitivity of 9.375 pg/mL, human proenkephalin (A) kit’s range of 78.125–5000 pg/mL and sensitivity of 46.875 pg/mL. The samples were defrosted slowly. The first step was transfer of the samples from −80 centigrade to −20 centigrade for a night; later, they were thawed in water with ice. According to pre-test measurements of 10 random samples, no necessary sample dilution was established for all tests. 

The assay procedures were performed according to the manufacturer’s manuals using a Bio-Rad washer and plate reader (Hercules, CA, USA). According to the manufacturer, all of the tests used have high sensitivity and excellent specificity for the detection of the measured parameters without significant cross-reactivity or interference between the analytes and their analogues. 

The study protocol was approved by the Jagiellonian University Bioethical Committee (approval no. 1072.6120.238.2019) and informed consent was obtained from all patients’ legal guardians and all patients over 16 years of age enrolled in the study.

### 2.3. Statistical Analysis

The statistical analysis was performed by Statistica 13.5 software (TIBCO Software Inc., Palo Alto, CA, USA) and GraphPad Prism 8 (GraphPad Software, San Diego, CA, USA). The endogenous opioid peptide concentrations were expressed as median values and min–max whiskers. Normality was checked using the Shapiro–Wilk test in each group. The differences between the study groups were performed by multiple comparison Friedman test and Dunn’s post hoc analysis. The Kruskal–Wallis test and Dunn’s post hoc analysis were performed for comparisons between the study groups and the control group. The correlation analysis was performed using Spearman’s correlation rank test. The number of patients enrolled in the study was verified.

## 3. Results

### 3.1. Patients’ Characteristics

Fifty adolescents with newly diagnosed CD (29 males, 22 females, mean age 14.4 ± 2.0 years) were enrolled in the study. During the study, all CD patients were treated with exclusive enteral nutrition using a semi-elemental diet for 6 weeks and 5-aminosalicylate (5-ASA, mesalazine, mesalamine) orally at a dose of 50 mg/kg. No patients were treated with anti-TNF alpha therapy, systemic nor local steroids, or probiotics, and no opioid analgesia was used during the treatment of CD. The control group consisted of 39 healthy adolescents (12 males and 27 females, mean age 14.0 ± 2.4 years). 

### 3.2. β-Endorphin

In the study group of adolescents diagnosed with CD, the serum concentrations of β-endorphin did not differ statistically at the different analyzed phases of the disease. The median and quartiles 1 and 3, respectively, were 31.5 (25.3–40.9) pg/mL in CD1, 24.7 (18.9–31.5) pg/mL in CD2, and 24.7 (20.6–38.3) pg/mL in CD3. During Crohn’s disease, in addition to its stage, we observed significantly lower concentrations of β-endorphin compared to the concentration in the control group of 47.2 (34.4–57.5) pg/mL (*p* = 0.007 between CD1 and C and *p* < 0.0001 in both the CD2 and CD3 groups versus the control group (C)) (Figure 1). 

### 3.3. Proenkephalin (PENK)

The PENK concentration also did not show statistically significant changes during CD. The PENK concentrations (results presented as median and quartile 1 and 3 in the parentheses) were 457.5 (191.0–696.8) pg/mL (before the treatment (CD1)), 238.5 (101.0–651.5) pg/mL (during the treatment (CD2)), and 222.5 (130.3–480.0) pg/mL (in the remission state (CD3)). However, the results for all the studied time points in this group were significantly lower compared to the control group (C) result of 1161.0 (810.5–1450) pg/mL, *p* < 0.0001 in all comparisons to the control group (Figure 2). 

### 3.4. Big Dynorphin

The serum concentrations of big dynorphin were also not statistically significant different among the variables assessed in all CD groups. In the CD1 group, we observed 18.0 (10.9–27.3) pg/mL (median and quartile 1 and 3, respectively); in the CD2 group, we observed 15.2 (6.7–26.7) pg/mL; and in the CD3 group, we observed 17.3 (11.2–21.4) pg/mL. Statistically significant differences in big dynorphin concentration were observed in the CD2 and CD3 groups only as compared to the control group median of 22.3 (15.6–32.00) pg/mL, *p* = 0.0066 and *p* = 0.032, respectively (Figure 3).

Spearman’s analysis was carried out, and no significant correlations were found between the analyzed endo-opioids in the studied and control groups. 

## 4. Discussion

This study is the first study to comprehensively assess changes in the concentrations of β-endorphin, proenkephalin, and big dynorphin, over the course of Crohn’s disease.

### 4.1. β-Endorphin

The analgesic potency of β-endorphin is strong, a dozen times stronger than exogenous morphine [31]. It acts centrally and peripherally. The peripheral action is, among others, realized by T lymphocytes containing β-endorphin. As shown in the paper by Wiedermann et al., the concentration of endorphins in leukocytes in Crohn’s disease is diminished by 50% compared to healthy people [32]. However, a newer study by Kuroki et al. showed an increase in endorphin concentration in IBD [33]. The clue could be that Kuroki analyzed IBD patients without selection for Crohn’s disease and colitis ulcerosa. According to Wiedermann, the concentration of endorphins in the leukocytes of patients suffering from colitis was unaffected. Our study showed that, in each phase studied, the concentration of β-endorphin in adolescents with Crohn’s disease was lower than in the control group, and these results are convergent to the Wiedermann study. Unfortunately, there are no studies that have described the plasma concentration of β-endorphin during the course of CD in adolescents. For this reason, available research was used in the discussion.

### 4.2. Enkephalins

Proenkephalin (PENK, formerly known as proenkephalin A) is the main biological source of two main enkephalins: met-enkephalin and leu-enkephalin. Therefore, PENK can be used as a marker of the activity of the endogenous opioid system. The results of a study by Owczarek et al. reported a decreased concentration of enkephalins in the serum of adults with CD; a result that is consistent with our observations in adolescents [34]. The authors of the study suggested that the reduction in enkephalin may be the result of chronic inflammation and migration of leukocytes capable of synthesizing enkephalin to diseased sites, resulting in insufficient peripheral production. There is also another reason; according to Zatorski et al. the decreased concentration of enkephalins may be due to the increased activity of dipeptidyl peptidase 4 (DPP-IV), an enzyme present on the plasma membrane of most types of somatic cells. This enzyme is associated with the regulation of immune processes and is responsible for the degradation of intestinal peptides, including GLP-1 and GLP-2, and endo-opioids. Among patients in the acute phase of IBD, an increased expression of DPP-IV was observed, which may be responsible for a decreased concentration of endo-opioids during the disease. Normalization of DPP-IV expression in the remission state resulted in an observed increase in enkephalin concentration [35]. This result also suggests a potential therapeutic role for DPP-IV inhibitors in the treatment of IBD. Currently, drugs from this group are used successfully in the treatment of type II diabetes, as DPP-IV inactivates beneficial incretin hormones.

Enkephalins may play a very important role in the treatment of Crohn’s disease, which has been indicated by research conducted on racecadotril and its metabolite thiorphan. Racecadotril is an inhibitor of endopeptidase, an endorphin-metabolizing enzyme that is also known under the name enkephalinase. Racecadotril is used for the treatment of acute diarrhea in children. A meta-analysis by Eberlin et. al. showed high effectiveness of racecadotril in the treatment of diarrhea in children, compared to that of loperamide [36]. 

### 4.3. Big Dynorphin

Big dynorphin precursor molecules consist of dynorphin A and B that have nociceptive properties, probably through the activation of KOR. This study is the first study to describe changes in dynorphin levels in the course of CD. The concentrations of big dynorphin in younger patients with CD mimic those observed for enkephalin and endorphin. Lower concentrations of dynorphin, as well as enkephalin, in people with CD than in healthy people may be associated with increased expression of DPP-IV in the course of IBD.

### 4.4. Limitations of this Study

The relatively small test group and the control group as well as the high biological variability of the analyzed compounds cause the results obtained in this study to represent a large dispersion, which may have resulted in the statistical significance not being reached. Testing more patients to eliminate the random factor should be considered. 

### 4.5. Links to the Crohn’s Disease Course and Treatment

In Crohn’s disease, the severity of inflammatory processes correlates with the severity of the disease and clinical manifestation [8,37]. The results of routine laboratory tests show elevated levels of inflammatory markers, while scientific studies have shown decreased levels of endo-opioids [38]. A comparison of the median of the compounds tested, taking into consideration the phases of the disease, shows a continuous decrease in the concentrations of β-endorphin, proenkephalin, and big dynorphin, despite the phase of disease. These results could be related to the chronicity of the disease, its genetic source, and its tendency to relapse. Stimulating the secretion of endogenous opioids during all phases of the disease seems to be beneficial; however, such an amplification during exacerbation of CD may bring the best effects. Endo-opioids have analgesic and anti-inflammatory properties. Reducing the perceived pain can significantly improve the quality of life of patients. Conversely, reducing inflammation can reduce or eliminate some symptoms, including pain. A substitute for such a treatment is more convenient analogs of endo-opioids and agonists of opioid receptors.

One of the modern and still researched forms of pain treatment is the application of biased opioid ligands. The drugs in this group are characterized by a strong analgesic effect and are devoid of dangerous residual side effects and abuse potential. MOR- and KOR-biased agonists are among the promising directions of development. The MOR-biased agonist is based on blocking signal transduction from the receptor to the β-arrestin. In an animal model (mice), these agonists have been proven to have prolonged analgesic effects, reduced gastrointestinal and respiratory depressant effects, and tolerance of morphine. Selective KOR agonists can selectively activate G-protein signaling without phosphorylating p38α MAPK. Drugs in this group are devoid of side effects in the form of dysphoria, sedation, anxiety, and depression. However, it is more desirable and safer to stimulate endogenous opioid synthesis [39,40,41,42]. 

Among the complications of Crohn’s disease, nutritional deficiencies are particularly important. Therefore, proper enteral or parenteral nutrition is very important in the treatment of CD [43]. The endo-opioid system stimulates the hunger center in the brain, also increasing appetite [44]. The reduced concentrations of endo-opioids in the course of CD weakens the appetite; hence, the equalization of endo-opioid concentrations would also be beneficial for the proper nutritional state of patients.

Long-term and chronic diarrhea is a very bothersome symptom of CD for patients. In addition to the reduced quality of life associated with the need to use the toilet frequently, there is a problem due to the loss of minerals from the gastrointestinal tract. Many antidiarrheal drugs, such as loperamide, contain opioids which act on opioid receptors located in the gut and thus inhibit intestinal transit. However, frequent use of opioid antidiarrheal drugs in combination with opioid analgesics can cause constipation, which is also uncomfortable for patients [45]. Hence, stimulation of endogenous opioid secretion may reduce the need for antidiarrheal drugs. 

The reduced concentrations of endo-opioids in CD can cause a general deterioration in mood, so the attitude toward living with the disease can also be negative. There have also been reports that decreased endo-opioid concentrations may cause or facilitate the appearance of depressive disorders [46]. At the same time, it has been confirmed that an increase in the concentration of β-endorphin was beneficial in the treatment of depression [47,48,49]. Stimulating the secretion of endogenous opioids can significantly improve a patient’s mood and general well-being.

In conclusion, in the course of CD, the endogenous opioid system is depressed, which promotes the development of inflammation, increases the perception of pain, and reduces appetite and general well-being. Equalizing endo-opioid concentrations should be part of the CD treatment strategy. Stimulation of endogenous opioid secretion can bring real benefits to patients in terms of alleviating disease symptoms, improving quality of life, and extending periods of remission, without the side effects associated with exogenous opioid use. Importantly, endogenous and exogenous opioids work using the same receptors located throughout the digestive tract. In the course of inflammatory bowel disease (IBD), the expression of opioid receptors in tissues increases, which makes them more susceptible to opioids [50]. 

The use of exogenous opioids often arouses numerous emotions related to the fact that they belong to drugs (narcotics). Therefore, developing therapies that increase the concentrations of endogenous opioids is recommended. This type of therapy should be widely accepted and, in addition, be free of numerous drug-related side effects such as addiction. In addition, the therapeutic effect of therapy modulating plasma levels of endo-opioids may be multidirectional—from their analgesic effect, through antidiarrhea, increasing appetite and nutritional status to improving mood.

## 5. Conclusions

We observed decreased concentrations of all the analyzed endo-opioids in the course of CD which is, in part, in line with other studies; however, many aspects are missing. Future research based on actual knowledge of endogenous opioids in Crohn’s disease should focus on modification of endogenous opioid secretions rather than using exogenous administration. This strategy may increase the benefits of opioid activities and decrease their potential adverse effects, which are often connected with their prolonged usage. Biased antagonists are very promising for patients with CD. The possibility of using opiates without their narcotic and addictive properties is particularly desirable in children and adolescents.

## Figures and Tables

**Figure 1 biomedicines-11-02037-f001:**
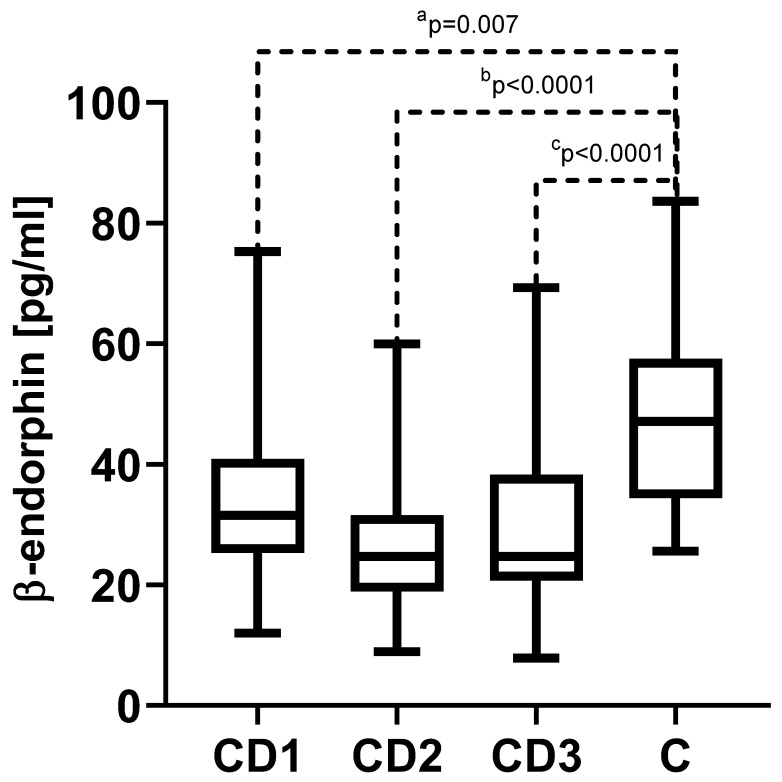
The measured serum concentrations of β-endorphin in Crohn’s disease (CD). CD1—Crohn’s disease in the acute phase; CD2—Crohn’s disease in the treatment phase; CD3—Crohn’s disease in the remission state; and C—control group; ^a^
*p*—statistical significance between CD1 and control group; ^b^
*p*—statistical significance between CD2 and control group; ^c^
*p*—statistical significance between CD1 and control group.

**Figure 2 biomedicines-11-02037-f002:**
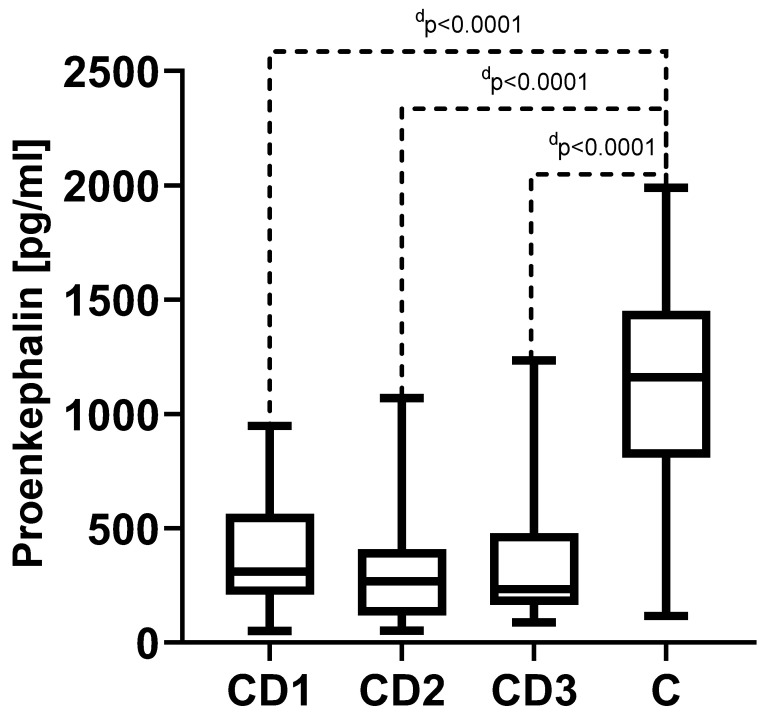
The measured serum concentrations of proenkephalin in Crohn’s disease (CD). CD1—Crohn’s disease in the acute phase; CD2—Crohn’s disease in the treatment phase; CD3—Crohn’s disease in the remission state; and C—control group. ^d^
*p*—statistical significance between CD1, CD2, CD3, and the control group.

**Figure 3 biomedicines-11-02037-f003:**
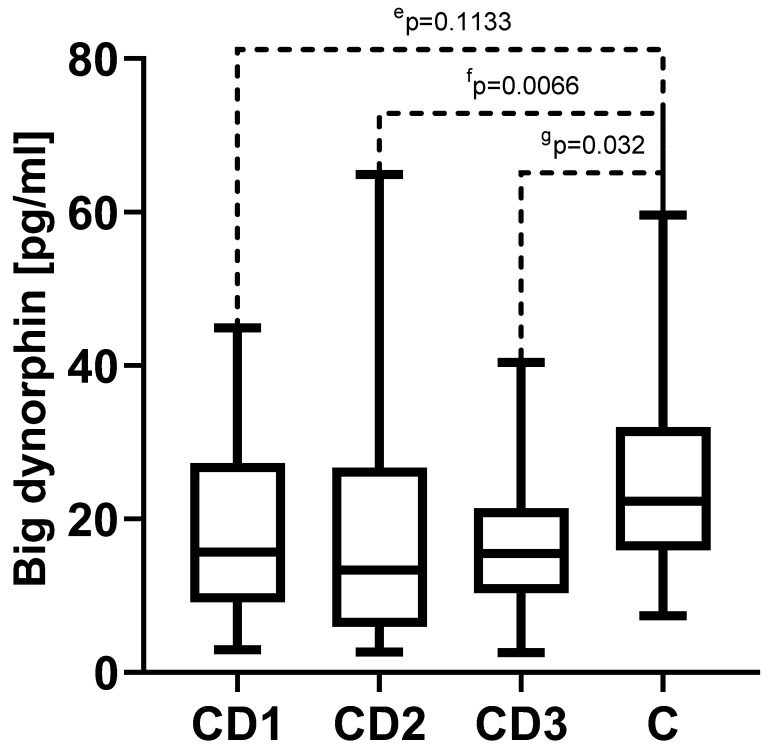
The measured serum concentrations of big dynorphin in Crohn’s disease (CD). CD1—Crohn’s disease in the acute phase; CD2—Crohn’s disease in the treatment phase; CD3—Crohn’s disease in the remission state; and C—control group; ^e^
*p*—statistical significance between CD1 and control group; ^f^
*p*—statistical significance between CD2 and control group; ^g^
*p*—statistical significance between CD3 and control group.

## Data Availability

The data presented in this study are available on request from the corresponding author. The data are not publicly available due to privacy.

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
