# Peer review of "Endogenous Opioids in Crohn’s Disease"

_biomedicines, 2023, doi:10.3390/biomedicines11072037_

Round 1
Reviewer 1 Report
Please provide literature data related to the classification of subjects 10 years to childhood and adolescence category.
Figure 3, fp: replace p= 0.066 with p<0.0066 (as indicated in section 3.3, p<0.0066)
Replace b with the symbol (beta)
Unify the nomenclature of the opioid receptors: κ-opioid receptor (KOR) vs kappa-opioid receptors (see the introduction and discussion sections)
Authors need to provide the full words stand for a-g abbreviations in figures 1-3.
Discussion: replace the levels of β-endorphin with the plasma levels of β-endorphin (Authors should specify the place of the changes in the endogenous opioid peptides).
Please incorporate thiorphan (a metabolite of racecadotril), an enkephalinase inhibitor in the discussion in relation to CD.
In relation to biased opioid ligands: Authors should also cite the original papers that provide data on the decrease in the side effects of MOR and KOR agonists.
Author Response
Reviewer 1.
Sir / Madam, Thank you very much for your valuable comments.
Please provide literature data related to the classification of subjects 10 years to childhood and adolescence category.
We corrected article due to WHO guidelines (https://www.who.int/health-topics/adolescent-health#tab=tab_1)
Figure 3, fp: replace p= 0.066 with p<0.0066 (as indicated in section 3.3, p<0.0066)
Corrected.
Replace b with the symbol (beta)
Corrected.
Unify the nomenclature of the opioid receptors: κ-opioid receptor (KOR) vs kappa-opioid receptors (see the introduction and discussion sections)
Corrected.
Authors need to provide the full words stand for a-g abbreviations in figures 1-3.
Corrected.
Discussion: replace the levels of β-endorphin with the plasma levels of β-endorphin (Authors should specify the place of the changes in the endogenous opioid peptides).
Unfortunately, there are no studies, that determine the concentration of b-endorphin in plasma in Crohn's disease. We used available studies, These limitations are described in the article.
Please incorporate thiorphan (a metabolite of racecadotril), an enkephalinase inhibitor in the discussion in relation to CD.
We added a paragraph in section 4.2.
In relation to biased opioid ligands: Authors should also cite the original papers that provide data on the decrease in the side effects of MOR and KOR agonists.
Thank you for this suggestion, we added a reference 39, 40, 41.
Reviewer 2 Report
- - Define the abbreviations the first time they appear in the full text (for example, “CD”) and not some paragraphers after. Then always use the abbreviation
- -“autoimmunological”
CD is not an autoimmune disease but an immune-mediated disease
- -Do not report number and baseline characteristics of included patients in the Methods but in the Results
- -“The control group consisted of 39 healthy children (12 boys and 27 girls, mean age: 96 14,0 ± 2,4 years) without any pathological clinical signs and complaints and any pharmacological treatment. In this group, fasting blood samples were drawn once.”
-How did you recruit controls?
- -“The number of patients enrolled in the study was verified. Twice more patients were enrolled in the study than the calculated minimum sample size.”
Explicit the calculation of the sample size
- - “This is the first study to comprehensively assess changes in the levels …” “colitis ulcerosa”
-Correct the English and revised the English in the whole text
- -“β-. endorphin” “. β-endorphin”
-Check the point. In addition, sometimes you wrote beta, sometimes b in italics, sometimes non in italics
- -Change the conclusions. Conclusion must be a data the directly derives from the results of your study and not a speculation
Must be revised.
Author Response
Reviewer 2.
Sir / Madam, Thank you for your kind review and notices.
Define the abbreviations the first time they appear in the full text (for example, “CD”) and not some paragraphers after. Then always use the abbreviation
Corrected.
autoimmunological” CD is not an autoimmune disease but an immune-mediated disease
Corrected.
Do not report number and baseline characteristics of included patients in the Methods but in the Results
Improved.
“The control group consisted of 39 healthy children (12 boys and 27 girls, mean age: 96 14,0 ± 2,4 years) without any pathological clinical signs and complaints and any pharmacological treatment. In this group, fasting blood samples were drawn once.” How did you recruit controls?
We added a description in paragraph 3.
The number of patients enrolled in the study was verified. Twice more patients were enrolled in the study than the calculated minimum sample size.”Explicit the calculation of the sample size
We used a statistic formula Nmin=NP(α2⋅f(1−f))NP⋅e2+α2⋅f(1−f), but we decided to delete this paragraph form article.
“This is the first study to comprehensively assess changes in the levels …” “colitis ulcerosa”Correct the English and revised the English in the whole text
English was corrected.
“β-. endorphin” “. β-endorphin” Check the point. In addition, sometimes you wrote beta, sometimes b in italics, sometimes non in italics
Checked and corrected, but β in subsection must be in italic.
Change the conclusions. Conclusion must be a data the directly derives from the results of your study and not a speculation
Thank you for the suggestion. We change the conclusion and add a few more information.
Round 2
Reviewer 2 Report
Thank you for the corrections.
The English has been improved.